# Activated *Ailanthus altissima* Sawdust as Adsorbent for Removal of Acid Yellow 29 from Wastewater: Kinetics Approach



Najeeb ur Rahman [1], Ihsan Ullah [1], Sultan Alam [1], Muhammad Sufaid Khan [1], Luqman Ali Shah [2], Ivar Zekker [3,*], Juris Burlakovs [4], Anna Kallistova [5], Nikolai Pimenov [5], Zane Vincevica-Gaile [6], Yahya Jani [7] and Mohammad Zahoor [8,*]

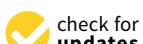



1   Department of Chemistry, University of Malakand, Chakdara Dir Lower, Chakdara 18800, Pakistan; nrnajeeb@yahoo.com (N.u.R.); Ihsanchemist877@gmail.com (I.U.); dr.sultanalam@yahoo.com (S.A.); sufaidkhan1984@gmail.com (M.S.K.)
2   National Centre of Excellence in Physical Chemistry University of Peshawar, Peshawar 25000, Pakistan; Luqman_alisha@yahoo.com
3   Institute of Chemistry, University of Tartu, 14a Ravila St., 51014 Tartu, Estonia
4   Institute of Forestry and Rural Engineering, Estonian University of Life Sciences, 5 Kreutzwaldi St., 51014 Tartu, Estonia; Juris.burlakovs@emu.ee
5   Winogradsky Institute of Microbiology, Research Centre of Biotechnology of the Russian Academy of Sciences, Leninsky Prospect, 33, Building 2, 119071 Moscow, Russia; kallistoanna@mail.ru (A.K.); npimenov@mail.ru (N.P.)
6   Department of Environmental Science, University of Latvia, LV-1004 Riga, Latvia; zane.vincevica-gaile@lu.lv
7   Unit of Built Environment and Environmental Science, Department of Urban Studies, Malmö University, 211 19 Malmö, Sweden; yahya.jani@mau.se
8   Department of Biochemistry, University of Malakand, Chakdara Dir Lower, Chakdara 18800, Pakistan
*   Correspondence: ivar.zekker@ut.ee (I.Z.); mohammadzahoorus@yahoo.com (M.Z.)

**Abstract:** In this study, *Ailanthus altissima* sawdust was chemically activated and characterized by Scanning Electron Microscopy (SEM), Fourier Transform Infrared (FTIR), Energy Dispersive X rays (EDX), and surface area analyzer. The sawdust was used as an adsorbent for the removal of azo dye; Acid Yellow 29 (AY 29) from wastewater. Different kinetic and equilibrium models were used to calculate the adsorption parameters. Among the applied models, the more suitable model was Freundlich with maximum adsorption capacities of 9.464, 12.798, and 11.46 mg/g at 20 °C, 30 °C, and 40 °C respectively while $R^2$ values close to 1. Moreover, the kinetic data was best fitted in pseudo second order kinetic model with high $R^2$ values approaching to 1. Furthermore, adsorption thermodynamics parameters such as free energy, enthalpy, and entropy were calculated and the adsorption process was found to be exothermic with a value of $\Delta H° = -9.981$ KJ mol$^{-1}$, spontaneous that was concluded from $\Delta G°$ values which were negative ($-0.275$, $-3.422$, and $-6.171$ KJ mol$^{-1}$ at 20, 30, and 40 °C respectively). A positive entropy change $\Delta S°$ with a value of 0.0363 KJ mol$^{-1}$ indicated the increase disorder during adsorption process. It was concluded that the activated sawdust could be used as a suitable adsorbent for the removal of waste material, especially dyes from polluted waters.

**Keywords:** chemically activated sawdust; azo dyes; adsorption; kinetics; isotherm models

## 1. Introduction

Water is considered as a vital substance, and without water, sustaining life on earth is impossible. Apart from drinking, it is used for domestic activities in daily life, and should be in pure form. However, increasing industrialization is the demand of increasing population, which has polluted the potable water supplies in almost all parts of the world and nowadays it is nearly impossible to get pure water for domestic uses. The industries discharge wastewater, which go directly into clean water sources, containing many different

types of pollutants, due to which the fresh water becomes unfit for domestic use. Among the pollutants, dyes are the most common contaminants discharged in large quantities into water bodies, being released from textile industries [1–3].

The main sources of dyes pollution are industries which make dyes and those which use dyes and pigments for coloring different objects. According to an estimation the annual production of dyes is about $7 \times 10^{8}$ kg annually which are available in more than 10,000 different varieties worldwide [4,5]. During the process of adsorption dye is not adsorbed 100% on the fabrics i.e., about 10–15% of the dye remains un-adsorbed which are released into drainage lines and finally are discharged to the water bodies [4]. The presence of dyes in water even in very small amount less than 1 ppm is highly observable and objectionable. Artificial dyes pose a severe environmental threat; they can change the quality of water, and are often poisonous, causing allergic reactions [5]. Severe health complications are also caused due to dye intoxication like dysfunction of liver, disorders related to nervous system and kidneys, etc. [6,7]. Azo dyes are usually acidic (anionic) in nature. The presence of azo group (N=N) with aromatic rings is the characteristic property of this class of dye. These dyes contain organic sulphonic acid (organic acid dye having a sulfur atom), acid salts of sodium [8]. Among the dyes azo dyes and water-soluble reactive dyes are biologically non- degradable, being stable to light and heat. Azo dyes are used to color leather, cotton, and wool silk. They are the part of daily household products, such as shower gels, bubble bath, liquid soaps, multipurpose cleaners, dish washing liquids, shampoos, and perfumes [9].

The dyes containing wastewater are treated by different physical, chemical and biological methods [10]. Most of these methods have some drawbacks; on one side these methods have high investment, high operating cast, complication in treatment processes and sludge production, and some of them are not workable with a wide range of dyes [11]. Among the categorized methods, adsorption is more frequently used for the removal of both color bearing substances and uncolored organic contaminators. It is noticeable that the applicability of the adsorption procedure mostly depends upon the low-price and easy availability of a variety of adsorbents [12]. For this purpose, different types of low-cost agricultural waste materials have been converted into activated carbons and even used as such after chemical treatments. Sawdust, comprised of fine particles, considered a more suitable adsorbent due to its low cost and easy availability everywhere and is obtained from sawmills as a byproduct where a variety of woods are cut down for different purposes. Saw dust is residual wooden powder. It is also considered as miner fuel which is used for the warming of houses [13,14]. To best of our knowledge, the chemically activated saw dust of *Ailanthus altissima* has not been used as adsorbent before. The porous structure and small size of sawdust particles would offer a high surface area for the adsorption of dyes and other pollutants.

The purpose of the present study was to effectively utilize the waste biomass in form sawdust in reclamation of water. The low cost chemically activated saw dust of *Ailanthus altissima* was thus evaluated for its adsorptive capacity whereas Acid yellow 29 (AY29) was the tested pollutant used in the study. To provide scientific grounds to the study the modified adsorbent was characterized by Scanning Electron Microscopy (SEM), Fourier Transform Infrared (FTIR), and surface area analyzer whereas different kinetics and isotherm models were used to determine the numerical values of the adsorption parameters.

## 2. Materials and Methods

### 2.1. Acid Yellow 29 as Adsorbate and Activation of Sawdust

The chemical structure of AY 29 is given in Figure 1 while its physiochemical properties are illustrated in Table 1. The sawdust of selected plant was collected from local wood mill, washed thoroughly to remove dust and soil remaining, dried in oven, then converted into fine powders. The powder sample was leached with standard 0.2 N nitric acid and hydrochloric acid solutions with the ratio of 1:1 (prepared from concentrated nitric acid and hydrochloric acid having active concentration acids as 70% and 36%, respectively, and

were purchased from Merck, Germany) and allowed to stay for 24 h at room temperature with regular shaking. The acid treatment is generally applied to open pores in such biosorbents [15]. It was filtered and washed with double distilled water until became free from $Cl^-$ and $NO^{3-}$ ions. The saw dust thus obtained was then air-dried in an oven at $105 \pm 2\,°C$.

**Figure 1.** Chemical structure of AY 29.

**Table 1.** Properties of the AY 29.

| Chemical Name | Acid Yellow 29 |
|---|---|
| Molecular Formula | $C_{22}H_{17}ClN_5NaO_6S_2$ |
| Molecular Weight | 569.97 g/mol |
| Classification | Anionic |
| Physical Description | Yellow Powder |
| $\lambda_{max}$ | 407 nm |
| Solubility in water | Concrete value of solubility in water |

### 2.2. Instruments Used in the Study

Sophisticated instruments were used while performing SEM (scanning electron microscopy), XRD (X-ray diffraction), EDX (Energy-dispersive X-ray spectroscopy), and BET (Brunauer, Emmet, and Teller) surface area analysis.

#### 2.2.1. UV-Spectrophotometer

To estimate concentration of dye in solutions, UV (ultraviolet and visible) spectrophotometer (UV-1800) was used. The dye stock solution was prepared by dissolving 0.0534 g of AY 29 in 1000 mL of distilled water in conical flasks, having effective concentration as 0.0001 M. Working dilutions were prepared using dilution formula. The absorbance of solutions before and after adsorbent applications was measured at 407 nm.

#### 2.2.2. SEM

The prepared adsorbent samples were analyzed by the SEM with specifications (JSM5910 JEOL, Tokyo, Japan), Manufacturer: JEOL, Japan, Energy: 30 KV, Magnification (Max): 300,000×, Resolving power (Max): 2.3 nm.

#### 2.2.3. XRD and EDX

The structural properties of the samples have been studied by XRD, with detail of X-ray Diffractometer Model: JDX-3532, Make:

JEOL, Japan, Voltage: 20–40 kV, Current: 2.5–30 mA, X-Rays: CuKα (λ = 0.154 nm), 2θ Range: 0 to 160°. Similarly, the EDX study was performed on INCA200 (Oxford instruments, Oxford, UK) to determine the elemental composition of the adsorbent.

2.2.4. BET Surface Area Analyzer

The BET surface area was analyzed by surface area analyzer, Model: NOVA2200e, Make: Quantachrome, Boynton Beach, FL, USA.

*2.3. Adsorption Experiments*

To determine the kinetics of selected dye adsorption two concentrations of 0.001M and 0.002 M were contacted with 0.1 g of adsorbent in 10 mL volume in a number of flasks. The flask were placed on a shaker and after a certain fixed interval, individual flask were taken out from shaker. Spectrophotometric analyses were carried out for remaining concentration to determine adsorption. The shaking was continued for 8 h. Pseudo first order and second order models were applied to decide best model to explain kinetics data. These models were used to measure the adsorption with respect to time at constant concentration and dissemination of adsorbate into the openings of the adsorbent.

To determine adsorption isotherm parameters 0.001, 0.002, 0.003, 0.004, 0.005, 0.006, and 0.007 M solutions in 10 mL volume were contacted with 0.1 g of adsorbent and shaken for 30 min (equilibrium time decided from above experiment). Different isotherm models were applied to get the best model that fits the equilibrium data.

In order to check the effect of temperature on the adsorption process, fixed concentration solutions in 10 mL were put in reaction with 0.1 g of sawdust and shaken at 20, 30, and 40°C. The obtained data was converted into Van't Hoff plot and the changes of different thermodynamic parameters, like enthalpy, entropy, and Gibbs free energy, were determined.

## 3. Results and Discussion
*3.1. Characterization of the Sample*
3.1.1. SEM Study

Using the scanning electron microscopy technique, characteristic topographies and morphologies of the sawdust were studied and examined. Figure 2a–d shows the SEM images of the sample at various magnifications. The sawdust surfaces have irregular shaped pores for the adsorption of adsorbate, which are clearly observable in the images.

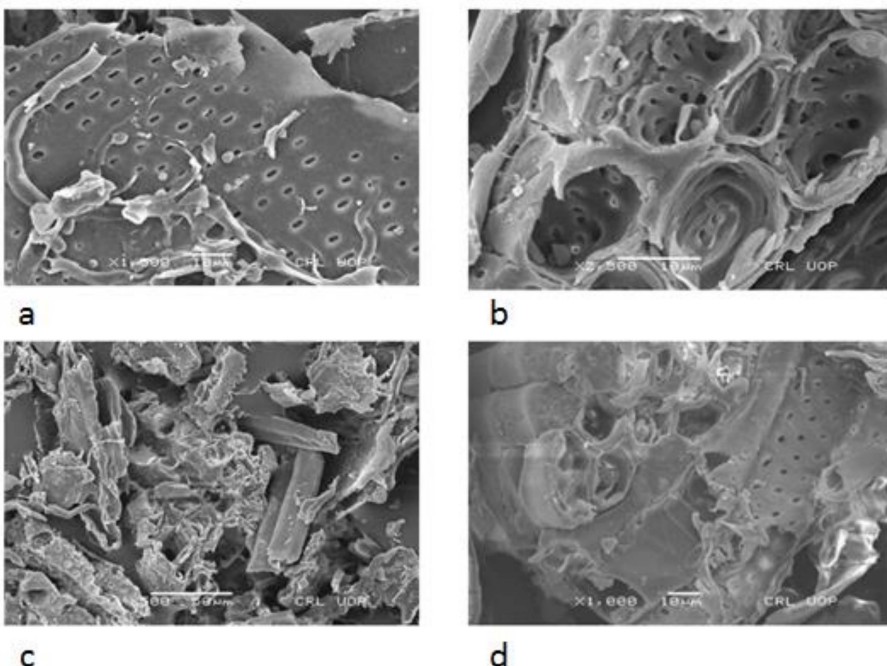

**Figure 2.** SEM images of sawdust at various magnifications (**a**) ×1000, (**b**) ×2500, (**c**) ×50 and (**d**) ×1000.

### 3.1.2. EDX Analysis

Figure 3 illustrates the EDX analysis for the activated sawdust consisted of a high percentage of carbon, nitrogen, and calcium. It is also shown that the surface has oxygen that binds to carbon in form of functional groups like aldehydes, ketones, and esters. The EDX plot also revealed the existence of potassium in small amounts as well.

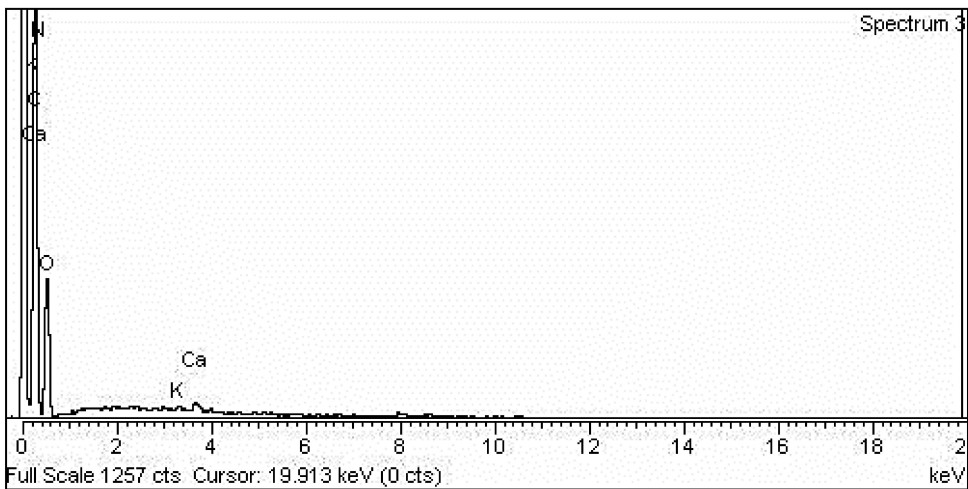

**Figure 3.** EDX spectrum of activated sawdust.

### 3.1.3. FTIR Analysis

The FTIR analysis (Figure 4) of carbonaceous adsorbents give knowledge about the functional groups that are responsible for the attachment of adsorbates to its surface and can be inferred from the existence of distinctive frequencies. Figure 4 shows FTIR spectrum of activated sawdust sample. The peak at 3327 cm$^{-1}$ provides details on the availability of the hydroxyl group (O-H) that may be a part of carboxylic acids, alcohols and phenols present in cellulosic fibers, lignin and pectins. Peak at 2880 cm$^{-1}$ indicates CH$_3$ stretching. The peak at 1728 cm$^{-1}$ shows the stretching vibration of carboxylic acid and ester C-O bonds [15]. The peak at 1582 cm$^{-1}$ represents COO-deprotonated carboxylate group. The peak observed at 1328 cm$^{-1}$ suggests the stretching vibrations of pectin (-COOH) [16]. The obtained peak at 1233 cm$^{-1}$ shows hemicellulose C-O stretching vibration and suggests the importation of hemicellulose C-O vibration. The presence of halogen group C-X on sawdust is suggested by the peak observed at 1035 and 535 cm$^{-1}$ [17].

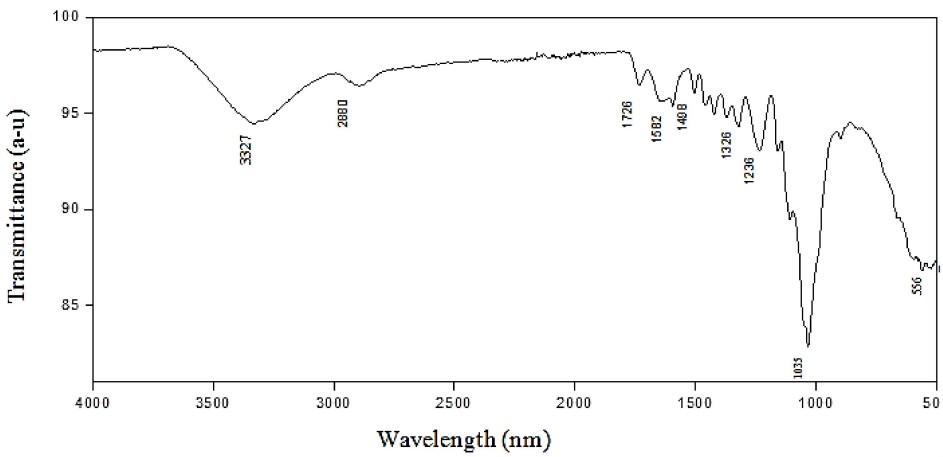

**Figure 4.** FTIR spectrum of sawdust sample.

### 3.1.4. Surface Area

The sawdust surface was determined using surface area analyser and surface area was found to be 279.646 m$^2$/g.

### 3.2. Kinetics Studies

### 3.2.1. The Effect of Contact Time on Adsorption

Figure 5 demonstrate the effect of the contact time of adsorbate over the surface of the adsorbent. Initially, the rate of adsorption increased very rapidly until around 30 min. After 30 min, the rate of the adsorption remained constant and no further increase in adsorption was observed because of the coverage of the surface of adsorbent by dye molecules. The time 30 min was, therefore, taken as the equilibrium time.

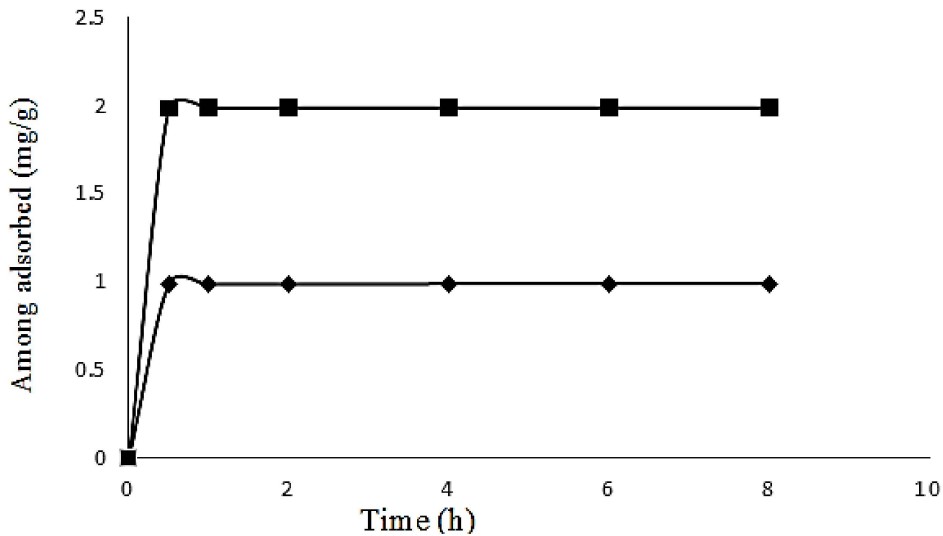

**Figure 5.** Equilibration time plot after adsorption AY 29 on sawdust.

### 3.2.2. Pseudo-1st Order Reaction Rate Equation

For understanding the kinetics of the adsorption of adsorbate (dye) on the adsorbent, the concept of Lagergren or pseudo first order kinetics model was used. The following straight line Equation (1) was used to determine kinetics constants [16–18]:

$$\frac{dCd\phi}{dt} = k1 \, (Cde - Cd\phi) \tag{1}$$

where dt is change in time, $C_{d\phi}$ is the amount of adsorbed (mg/g) dye on the adsorbent (sawdust), $C_{de}$ is the equilibrium potential of adsorption in mg/g and $k_1$ is the rate constant/minutes. Moreover, logarithmic form of the equation can be given by Equation (2) [16–18]:

$$\ln(Cde - Cd\phi) = \ln(Cde - Cd\phi) \tag{2}$$

A graph was plotted between $\ln(Cde - Cd\phi)$ and t, from the slope of which k1 while from the intercept Cde were calculated and presented in Figure 6a.

### 3.2.3. Pseudo Second Order Kinetic Equation

Mathematically, this model can be given as follows [16–18]:

$$\frac{T}{C_{d\phi}} = \frac{1}{C_{de}} T + \frac{1}{k_2 \, C_{de}^2} \tag{3}$$

where $C_{de}$ corresponds to the equilibrium adsorption potential of sawdust, $C_{d\phi}$ is the adsorption potential of the adsorbent (sawdust), and $K_2$ is the rate constant. A plot of $\frac{T}{C_{dO}}$ vs t gives us a graph with a slope of $\frac{1}{C_{de}}$ and intercept $\left(\frac{1}{k_2C_{de}^2}\right)$ as given in Figure 6b.

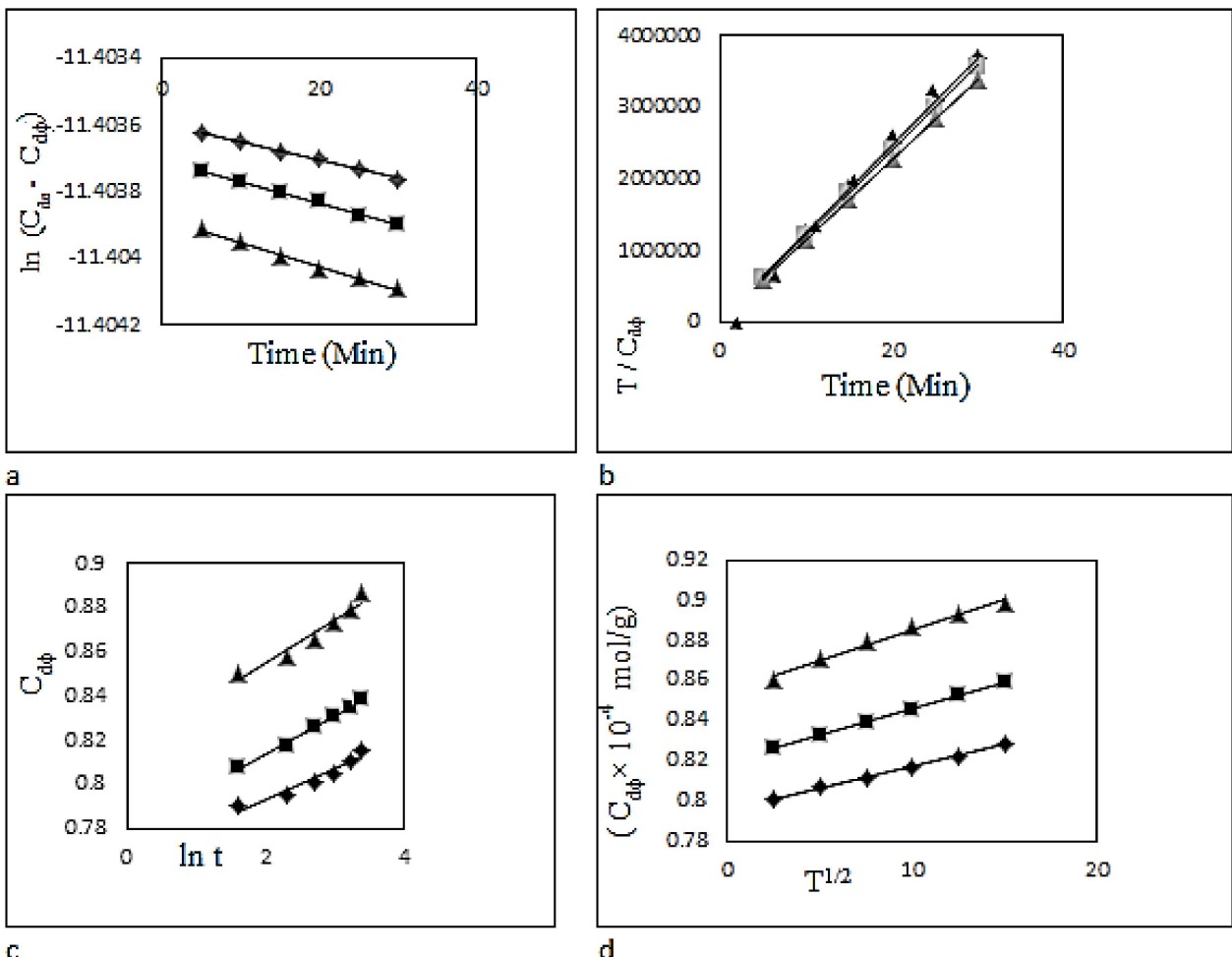

**Figure 6.** Adsorption of AY 29 on sawdust (**a**) Plot of pseudo 1st order rate constant. (**b**) Plot of pseudo 2nd order rate constant (**c**) Elovich plot (**d**) Kinetic model of intraparticle diffusion.

### 3.2.4. Elovich Model

The Elovich model's mathematical representation ca be given as follows [16–18]:

$$C_{d\phi} = \frac{1}{A}\ln(B\,A) + \frac{1}{A}\ln(t) \tag{4}$$

A graph of $C_{d\phi}$ and lnt is given Figure 6c where $1/A$ is a slope while $1/A\ln(BA)$ is an intercept.

### 3.2.5. Intraparticle Diffusion Model

This model is used to explain the adsorption phenomenon of kinetics of adsorption that basically determine the rate limiting step. It can be given mathematically as follows [16–18]:

$$C_{d\phi} = k_{ip}\,t_{\frac{1}{2}} + C \tag{5}$$

where $k_{ip}$ is diffusion rate constant and C is the thickness of the liquid film that can be obtained from intercept of $C_{d\phi} \times 10^{-4}$ vs $t_{/2}$ plot Figure 6d. The straight-line of the graph shows that the adsorption process is a directly controlled by diffusion and if it does not pass by the center in with many linear sections, it may suggest that boundary diffusion is not the only step involved as a mechanism. The values of calculated parameters are shown in Table 2

**Table 2.** Comparison data for the parameters of the various kinetic equations following the adsorption of AY 29 on sawdust.

| Kinetic Model | Parameter | Adsorption Temperatures | | |
|---|---|---|---|---|
| | | 10°C | 20°C | 30°C |
| Pseudo-first order | qe | 9.467 | 12.798 | 11.46 |
| | k1 | $-5 \times 10^{-6}$ | $-6 \times 10^{-6}$ | $-7 \times 10^{-6}$ |
| | $R^2$ | 0.9974 | 0.9977 | 0.99 |
| Pseudo-second order | qe | 0.001792 | 0.003306 | 0.001947 |
| | k2 | 121876 | 111880 | 118227 |
| | $R^2$ | 0.9999 | 1 | 0.9999 |
| Intra particle diffusion | $k_{ip}$ | 0.002 | 0.0024 | 0.0028 |
| | Intercept | 0.7855 | 0.8052 | 0.8442 |
| | $R^2$ | 0.9972 | 0.9991 | 0.9865 |
| Elovich | B | 0.0136 | 0.0172 | 0.0195 |
| | $R^2$ | 0.9376 | 0.9937 | 0.9503 |

*3.3. Isotherm Studies*

3.3.1. Langmuir Adsorption Model

The mathematical description of this model is as follows [16–18]:

$$\frac{Ce}{qe} = \frac{Ce}{qm} + \frac{1}{KLqm} \tag{6}$$

where $q_e$ is equilibrium concentration and Ce = Dye concentration

Based on the variance and difference in the related surface area and the adsorbent surface pore volume, $K_L$ = Langmuir constant interlinked with removal efficiency, which means that the greater and maximum adsorption potential is due to the huge surface area and massive amount of pores. If the adsorption process follows the isothermal model of Langmuir, the graph constructed in Figure 7a–c has a straight line (between $C_e/q_e$ and $C_e$). The values of the constants are given in Table 3 along with $R^2$ values.

**Table 3.** Comparison of the different parameters of the various isothermal models.

| Isotherm Models | Parameters | Adsorption Temperatures | | |
|---|---|---|---|---|
| | | 20 °C | 30 °C | 40 °C |
| Langmuir | $q_{max}$ | 0.8095 | 0.4299 | 0.7457 |
| | $K_L$ | 1.9462 | 1.0834 | 0.4235 |
| | $R^2$ | 0.8451 | 0.7231 | 0.8433 |
| Freundlich | slope | 0.6985 | 0.4896 | 11.46 |
| | $k_f$ | 0.0629 | 0.2417 | 0.7549 |
| | $R^2$ | 0.9654 | 0.8865 | 0.1676 |
| Temkin | B1 | 0.1619 | 0.2858 | 0.1321 |
| | KT | 0.2024 | 0.9376 | 0.5149 |
| | $R^2$ | 0.8891 | 0.7435 | 0.7312 |

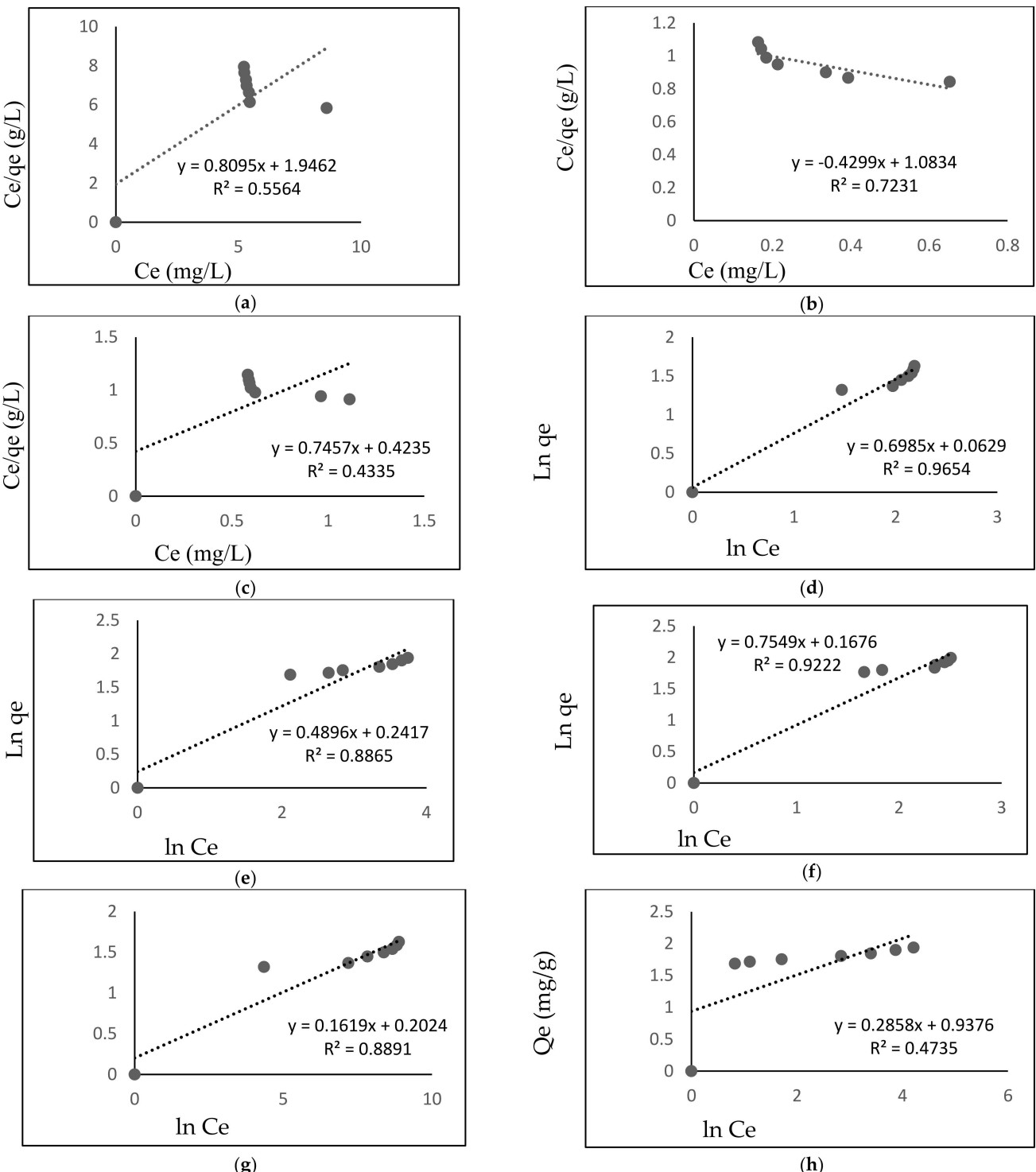

**Figure 7.** Langmuir Adsorption Isotherm for AY 29 on Sawdust at (**a**) 20 °C (**b**) 30 °C (**c**) 40 °C, Freundlich Adsorption Isotherm (**d**) 20 °C (**e**) 30 °C (**f**) 40 °C, Temkin Adsorption Isotherm (**g**) 20 °C (**h**) 30 °C.

### 3.3.2. Freundlich Adsorption Isotherm Model

This empirical equation is used to explain adsorption of adsorbates on the adsorbent surface where multilayer formation is predominant. The mathematical form of this model is given below [16–18]:

$$\ln q_e = \ln K_i + \frac{1}{n} \ln C_e \tag{7}$$

were qe is the amount adsorbed at equilibrium, $C_e$ is remaining concentration of dye at equilibrium. $K_f$ is Freundlich constant, and $1/n$ is the empirical constant. From the plot $\ln q_e$ vs. $\ln C_e$, the $1/n$ and $K_i$ in Figure 7d–f was determined from slope and intercept.

### 3.3.3. Temkin Isotherm Model for Adsorption

This adsorption isotherm model is commonly used for gas phase adsorption and deduces that with the adsorbent saturation the adsorption energy sequentially decreases [18]. The linear form of this model can be given as follows [16–18]:

$$qe = B1 \ln KT + B1 \ln Ce \tag{8}$$

where B1 and KT are adsorption and equilibrium binding constants and their values can be estimated from the slope and intercept of $q_e$ vs. $\ln C_e$ plot.

The Temkin isothermal adsorption of acid yellow 29 on sawdust is shown in Figure 7g,h while the calculated constants values are given in Table 4.

**Table 4.** Thermodynamic parameters of the adsorption of AY 29 on sawdust.

| $\Delta G^\circ$ (KJmol$^{-1}$) | | | $\Delta H^\circ$ (KJmol$^{-1}$) | $\Delta S^\circ$ (KJ mol$^{-1}$K$^{-1}$) | Ea (KJmol$^{-1}$) |
|---|---|---|---|---|---|
| 20 °C | 30 °C | 40 °C | −9.981 | 0.0363 | 0.013 |
| −0.275 | −3.422 | −6.171 | | | |

Based on $R^2$ values, Freundlich model is the best to explain isothermal data of adsorption.

### 3.4. Thermodynamics Parametes for Adsorption of AY 29 on Sawdust

Thermodynamics parameters such as free energy $\Delta G^\circ$, enthalpy $\Delta H^\circ$, and entropy $\Delta S^\circ$ were calculated using the following equations [16–18]:

$$\Delta G = -RT \ln K_D \tag{9}$$

$$K_D = \frac{q_e}{C_e}. \tag{10}$$

$$\Delta G = \Delta H - T\Delta S \tag{11}$$

Therefore,

$$\Delta H - T\Delta S = -RT \ln K_\circ \tag{12}$$

or,

$$\ln K_\circ \cdot \frac{\Delta S}{R} - \frac{\Delta H}{RT} \tag{13}$$

where $K_0$ is the adsorption distribution coefficient, $\Delta G^\circ$ (KJmol$^{-1}$) is the free energy of adsorption, $T$ (Kelvin) is the absolute temperature, $R$ is the universal gas constant, $\Delta H^\circ$ (KJmol$^{-1}$) is the heat of adsorption, and $\Delta S$ (KJmol$^{-1}$K$^{-1}$) is the entropy change associated with the adsorption of adsorbate on the adsorbent surface.

Figure 8 indicates the plots of $\ln K$ versus $1/T$ for the adsorption of AY 29 from the slope of which enthalpy $\Delta H^\circ$ and from intercept entropy $\Delta S^\circ$ values have been estimated [15,19].

Table 4 shows the estimated thermodynamics parameters where $\Delta G^\circ$ is negative, indicating the feasibility and spontaneity of the process. The negative values of $\Delta H^\circ$ shows the exothermic nature of the process. The positive values of entropy $\Delta S^\circ$ shows the increase disorder at the interfaces during adsorption process [20–22]. Figure 8 represents Van't Hoff graph for AY 29.

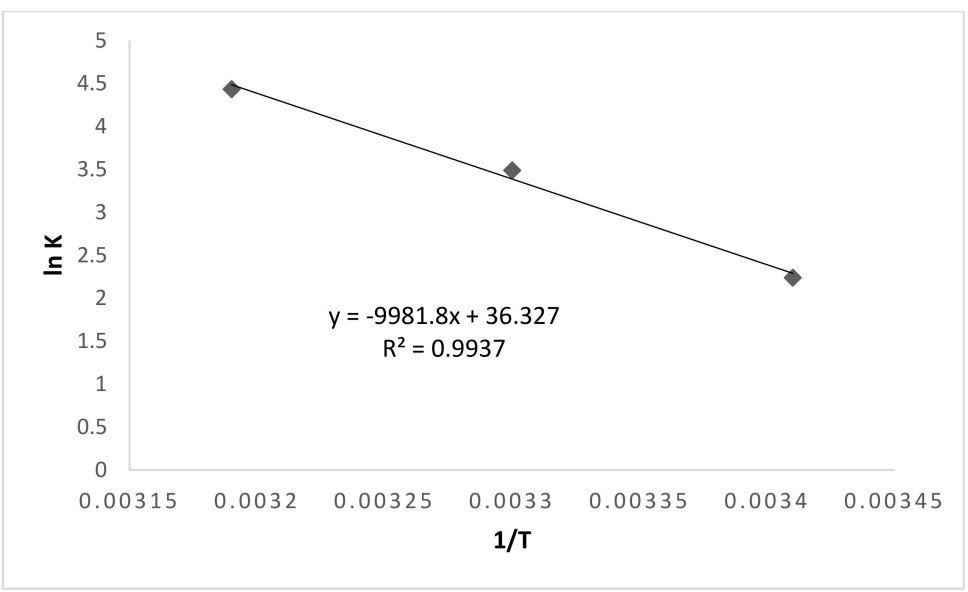

**Figure 8.** Van't Hoff graph for AY 29.

*3.5. Comparison of Adsorption Capacities of Present Adsorbent with Those Reported in Literature*

Sawdust as such has not been used as an adsorbent. In the literature, it has mostly been converted into activated carbon. A comparison of present adsorbent capacity with the reported one in literature is given in Table 5. However, one must not be confused about its low adsorbent capacity value as activated carbon being used in the cited studies have a high surface area as compared to our present adsorbent [23–25].

**Table 5.** Comparison of adsorption capacities of sawdust-based adsorbents for azo dyes with current research work.

| Dyes | Adsorbent | Qmax (mg/g) | Reference |
|---|---|---|---|
| Methylene blue and brilliant blue | Basic sawdust acacia | 8.13 and 267.04 | [23] |
| | Acidic sawdust acacia | 6.19 and 230.76 | |
| Basic Red 46 and Reactive Red 196 | sawdust-based adsorbent | 13.94 and 13.39 | [24] |
| Methylene blue | Rattan sawdust | 294.14 | [25] |
| Acid Yellow 29 | Activated Ailanthus altissima Sawdust | 13 | Present study |

The activated sawdust effectively removed the selected dye from water. The present adsorbent which is cost effective, has advantages over the reported activated carbon adsorbents, as in their preparation, pyrolysis in furnace needs high energy supply which from economical point of view is not a good sign. Activated *Ailanthus altissima* sawdust is a novel adsorbent and has not previously been used as an adsorbent for dye removal. However, further studies are needed to confirm its applicability and large-scale use for other pollutants.

**4. Conclusions**

The focus of the study was to devise an efficient adsorbent for the removal of AY 29. Activated *Ailanthus altissima* sawdust effectively removed the selected dye from aqueous solution with Langmuir as best model that fitted well equilibrium adsorption data while kinetics data was more effectively accommodated by the pseudo second order model. The thermodynamics feasibility of the process is evident from the negative values of Gibbs free energy and enthalpy change; moreover, the positive value of entropy indicated the favorable nature of the process. It was concluded from the results that activated sawdust could be effectively used as adsorbent for the removal of selected dyes and other pollutants.

**Author Contributions:** Conceptualization, N.u.R.; methodology, I.U.; software, S.A.; validation, I.Z., M.S.K. and M.Z.; formal analysis, A.K., J.B.; investigation, I.Z., A.K., N.P.; resources, M.Z.; data curation, L.A.S.; writing—original draft preparation, I.Z. and Y.J.; writing—review and editing, I.Z.; visualization, Z.V.-G.; supervision, M.Z.; project administration, M.Z.; funding acquisition, M.Z. All authors have read and agreed to the published version of the manuscript.

**Funding:** This research was funded by project nr T190087MIMV and European Commission, MLTKT19481R "Identifying best available technologies for decentralized wastewater treatment and resource recovery for India, SLTKT20427 "Sewage sludge treatment from heavy metals, emerging pollutants and recovery of metals by fungi and by project KIK 15392 and 15401 by European Commission.

**Institutional Review Board Statement:** Not applicable.

**Informed Consent Statement:** Not applicable.

**Data Availability Statement:** The data associated in this publication is totally presented in this paper. No data is present in any other repository anywhere.

**Conflicts of Interest:** The authors declare no conflict of interest. The funders had no role in the design of the study; in the collection, analyses, or interpretation of data; in the writing of the manuscript, or in the decision to publish the results.

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
