# Peer review of "Activated Ailanthus altissima Sawdust as Adsorbent for Removal of Acid Yellow 29 from Wastewater: Kinetics Approach"

_water, doi:10.3390/w13152136_

Round 1

Reviewer 1 Report

The topic of the paper is interesting where it seems to be suitable for the publication in Water Journal. However, before publishing I suggest to improve the text in accordance with my suggestions situated below.

Page 2, line 49: the period leading for the production of stated amount should be present.

Page 2, line 59: the term "organic sulphuric acid" is not quite clear for me. Is it possible to specify it? 

Page 2, line 64: "The dyes containing wastewaters" instead "The dyes containing effluents"?

Page 2, line 67: sludge instead slug?

Table 1: concrete value of the solubility in water instead the statement "Soluble in water"?

Figure 3: the style of the picture should be improved with the aim to increase the visibility of the particular elements symbols

Figure 6: the style should be improved in many ways (e.g. the presentation of the units for each axis, the placement of the curves on fig 6 b, positions of particular pictures etc.).

Other figures: please, check the position of the pictures in the text

Author Response

Reviewer 1

The topic of the paper is interesting where it seems to be suitable for the publication in Water Journal. However, before publishing I suggest to improve the text in accordance with my suggestions situated below.

  • Thank you worthy reviewer, for your positive input and attitude.

Page 2, line 49: the period leading for the production of stated amount should be present.

  • Worthy Reviewer, the required changes were made accordingly.

Page 2, line 59: the term "organic sulphuric acid" is not quite clear for me. Is it possible to specify it?

  • Worthy Reviewer, the organic sulphonic acid mean organic acid dyes containing sulphar atom.

Page 2, line 64: "The dyes containing wastewaters" instead "The dyes containing effluents"?

  • Worthy Reviewer, the suggested changes have been incorporated accordingly.

Page 2, line 67: sludge instead slug?

  • Worthy Reviewer, the suggested changes have been incorporated accordingly.

Table 1: concrete value of the solubility in water instead the statement "Soluble in water"?

  • Worthy Reviewer, The required changes have been incorporated accordingly.

Figure 3: the style of the picture should be improved with the aim to increase the visibility of the particular elements symbols

  • Worthy Reviewer, The improved picture was incorporated accordingly.

Figure 6: The style should be improved in many ways (e.g. the presentation of the units for each axis, the placement of the curves on fig 6 b, positions of particular pictures etc.).

  • Worthy Reviewer, The suggested changes have been incorporated accordingly.

Other figures: please, check the position of the pictures in the text

  • The figures were thoroughly checked and corrections were made where needed.

Reviewer 2 Report

The manuscript entitled “Activated Ailanthus altissima Sawdust as Adsorbent for Removal of Acid Yellow 29 from Wastewater: Kinetics Approach” presents an interesting research aimed at the assessment of the possibilities of using sawdust for water treatment.

The concept of this study is interesting and can be valuable contribution when considering practical applications of materials in environmental engineering.

Nevertheless, some amendments and clarifications are needed to increase the quality of the paper.

General comments:

  • Please explain all abbreviations that occur in the text. For example, in the abstract you mention SEM, FTIR, EDX, but it is required to give the full name in the place where the abbreviations are used for the first time.
  • Please use SI units in the manuscript. In line 49: “tons” are not used adequately.
  • The authors defined the problem of water contamination (especially by dyes) in reference to many scientific research, but in my opinion the research hypothesis and the objectives of the study are not highlighted. Also, the literature review must be done concerning more up to date research findings. In my opinion it is not necessary to cite publications from 1995, 1999 or 2000, but better focus on the newest data from the last 5 years.
  • The authors only stated that “In the present study low cost chemically activated saw dust of Ailanthus altissima was used for elimination of azo dyes (Acid yellow 29) (AY29) from waste water after chemical modifications. To provide scientific grounds to the study the modified adsorbent has been characterized by SEM, FTIR and surface area analyzer”. It is linked to the methodology, but what was the purpose of the study?
  • The methodology should be improved. Please cite the method described in lines 100-104.
  • Line 107: what is the specification JSM5910? It is not cited anywhere.
  • In chapter 2.2 the authors describe instruments, so I suggest to remove “studies” from 2.2.3 in line 109 and give the name of the apparatus.
  • The description of adsorption experiments in the chapter 2.3 should be amended. Please cite the procedure you followed in this analysis.
  • It is not explained why did the authors use pseudo first order and second order models for kinetic studies. There is also no link to the literature where these models are presented.
  • The quality of figures is very low. Please improve the resolution.
  • The discussion is poor. Please enhance the discussion and elaborate more about the outcomes of the study.
  • The content presented in lines 188-213 does not refer to the results. The description of model used should be placed in the chapter “Materials and methods”. Also, the references for each model are needed.
  • Figure 6 should be improved. The quality is not acceptable.
  • The isotherm models (lines 220-245) should be described in “Materials and methods”.
  • Figure 7 should be improved. The quality is not acceptable.
  • All tables and figures should be cited in the text. You cannot add the figure or the table to the manuscript with no comment on it.
  • The method of thermodynamic analysis should be described in the chapter” Material and methods”.
  • I expect to see the limitations of performed study in the Discussion and confrontation of obtained results with results presented by other researchers.
  • Please highlight what is the novelty in performed research.
  • The theme of the manuscript is valuable, nevertheless the content must be better arranged and improved. The manuscript must be thoroughly verified and major revisions are required before further processing of the paper.

Author Response

Reviewer 2:

The manuscript entitled “Activated Ailanthus altissima Sawdust as Adsorbent for Removal of Acid Yellow 29 from Wastewater: Kinetics Approach” presents an interesting research aimed at the assessment of the possibilities of using sawdust for water treatment.

The concept of this study is interesting and can be valuable contribution when considering practical applications of materials in environmental engineering.

  • Thank you worthy reviewer, for your positive input and efforts that you made in improving our research work.

Nevertheless, some amendments and clarifications are needed to increase the quality of the paper.

General comments:

  • Please explain all abbreviations that occur in the text. For example, in the abstract you mention SEM, FTIR, EDX, but it is required to give the full name in the place where the abbreviations are used for the first time.
  • Worthy Reviewer, they were elaborated in abstract as there was no further use in abstract therefore elaborated names were left while abbreviations were omitted.

  • Please use SI units in the manuscript. In line 49: “tons” are not used adequately.

  • Worthy Reviewer: they were accordingly corrected. .
  • The authors defined the problem of water contamination (especially by dyes) in reference to many scientific research, but in my opinion the research hypothesis and the objectives of the study are not highlighted. Also, the literature review must be done concerning more up to date research findings. In my opinion it is not necessary to cite publications from 1995, 1999 or 2000, but better focus on the newest data from the last 5 years.
  • Worthy Reviewer: the hypothesis and objective statement was added accordingly into introduction section. References were updated accordingly.
  • The authors only stated that “In the present study low cost chemically activated saw dust of Ailanthus altissima was used for elimination of azo dyes (Acid yellow 29) (AY29) from waste water after chemical modifications. To provide scientific grounds to the study the modified adsorbent has been characterized by SEM, FTIR and surface area analyzer”. It is linked to the methodology, but what was the purpose of the study?

  • Worthy Reviewer, the paragraph was rephrased in such a way to clarify purpose of the study effectively.
  • The methodology should be improved. Please cite the method described in lines 100-104.

Worthy Reviewer: The methodology has been improved and cited as well in the revised manuscript.

  • Line 107: what is the specification JSM5910? It is not cited anywhere.
  • Worthy Reviewer: Specifications have been incorporated in revised manuscript accordingly.
  • In chapter 2.2 the authors describe instruments, so I suggest to remove “studies” from 2.2.3 in line 109 and give the name of the apparatus.

Worthy Reviewer: it was removed accordingly.

  • The description of adsorption experiments in the chapter 2.3 should be amended. Please cite the procedure you followed in this analysis.

Worthy Reviewer: The suggestions was incorporated accordingly.

  • It is not explained why did the authors use pseudo first order and second order models for kinetic studies. There is also no link to the literature where these models are presented.
  • Worthy Reviewer: Pseudo first order and second order models were used to measure the kinetics parameters. The required link was accordingly provided in the revised manuscript in form of proper citation.
  • The quality of figures is very low. Please improve the resolution.

Worthy Reviewer: The quality of Figures has been improved.

  • The discussion is poor. Please enhance the discussion and elaborate more about the outcomes of the study.

Worthy Reviewer: We try our best and refined again the discussion accordingly.

  • The content presented in lines 188-213 does not refer to the results. The description of model used should be placed in the chapter “Materials and methods”. Also, the references for each model are needed.

Worthy Reviewer: Than for suggestions and placed in specified places accordingly..

  • Figure 6 should be improved. The quality is not acceptable.

Worthy Reviewer: The quality has been improved.

  • The isotherm models (lines 220-245) should be described in “Materials and methods”.

Worthy Reviewer: Sentences shifted to Materials and methods accordingly.

  • Figure 7 should be improved. The quality is not acceptable.

Worthy Reviewer: The quality has been improved.

  • All tables and figures should be cited in the text. You cannot add the figure or the table to the manuscript with no comment on it.

Worthy Reviewer: All Tables and Figures have been cited in the text accordingly.

  • The method of thermodynamic analysis should be described in the chapter” Material and methods”.

Worthy Reviewer: Has been described in ” Material and methods”.

  • I expect to see the limitations of performed study in the Discussion and confrontation of obtained results with results presented by other researchers.
  • Worthy Reviewer in last paragraph of discussion section required details have been incorporated accordingly. A comparison with other adsorbents has been provided in section 3.5 and table 4. Hope it will now be OK.
  • Please highlight what is the novelty in performed research.
  • Worthy Reviewer, the Ailanthus altissima sawdust in activated form has not been used before as adsorbent for dye or other pollutant removal from water therefore we claim that it is a novel adsorbent. This statement has been added to introduction section accordingly.
  • The theme of the manuscript is valuable, nevertheless the content must be better arranged and improved. The manuscript must be thoroughly verified and major revisions are required before further processing of the paper.

Thank you worthy reviewer, we have thoroughly revised the paper according to your worthy suggestions. Hopefully, it will be ok now. 

Round 2

Reviewer 2 Report

The authors have improved the manuscript, nevertheless some issues still have to be amended. 

The abbreaviations should be explained in the place where you used them for the first time. For example, you mentioned SEM in line 87, but you explained it in line 107.  The abbreviation of FTIR was not explained.

The font of "Energy-dispersive X-ray spectroscopy" in line 108 is different from the rest of the text.

Please focus on the quality of figures. Figure 2 is not acceptable? What is the purpose of the yellow frame on this article? Why is the "lnk" underlined on the vertical axis?

Why are "a", "b", "c" and "d" underlined in Figure 3? The scale in figure 3 is not visible.

The frame is not necessary in Figure 5. The should be also "Figure 5" in the description instead of "FiFigure 5".

Please remove the frame from Figure 6.

Please correct the Figure 7. The font size in vertical and horizontal axes should be the same in each graph. 

Why is the "Power function kinetic model" bolded in Table 2?

Figure 8 is not acceptable. Please align the location of the graphs on the page. Now they are placed disorderly.

The font in Table 4 is different from the rest of the manuscript.

Please correct the style of references used. All new publications added are not consistent with the requirements of the journal.

Author Response

Reviewer comments

We are extremely thankful to the honourable reviewers who thoroughly reviewed our work and presented their valuable comments for our guidance.  

Reviewer #1: The abbreviations should be explained in the place where you used them for the first time. For example, you mentioned SEM in line 87, but you explained it in line 107.  The abbreviation of FTIR was not explained.

  • Worthy reviewer, the abbreviations were expanded accordingly.

The font of "Energy-dispersive X-ray spectroscopy" in line 108 is different from the rest of the text.

  • Worthy reviewer, the font size has been corrected accordingly.

Please focus on the quality of figures. Figure 2 is not acceptable? What is the purpose of the yellow frame on this article? Why is the "lnk" underlined on the vertical axis?

  • Worthy reviewer, Figure 2 has been corrected with removal of yellow frame etc.

Why are "a", "b", "c" and "d" underlined in Figure 3? The scale in figure 3 is not visible.

  • Worthy reviewer, has been removed underline from a, b, c and d Figure 3 and enlarge the scale.

The frame is not necessary in Figure 5. The should be also "Figure 5" in the description instead of "FiFigure 5".

  • Worthy reviewer, the frame has been removed and corrected accordingly.

Please remove the frame from Figure 6.

  • Worthy reviewer, the frame has been removed from Figure 6 accordingly.

Please correct the Figure 7. The font size in vertical and horizontal axes should be the same in each graph. 

  • Worthy reviewer, the font size in vertical and horizontal axes revised and now the same in each graph. 

Why is the "Power function kinetic model" bolded in Table 2?

  • Worthy reviewer, "Power function kinetic model" bolded in Table 2 converted into nomal.

Figure 8 is not acceptable. Please align the location of the graphs on the page. Now they are placed disorderly.

  • Worthy reviewer, Figure has been corrected and placed in order accordingly.

The font in Table 4 is different from the rest of the manuscript.

  • Worthy reviewer, the font of Table 4 now similar to the rest of manuscript.

Please correct the style of references used. All new publications added are not consistent with the requirements of the journal.

  • Worthy reviewer, the new added references style has been corrected accordingly.